# Internal and External Demands in Basketball Referees during the U-16 European Women’s Championship

**DOI:** 10.3390/ijerph16183421

**Published:** 2019-09-14

**Authors:** David García-Santos, José Pino-Ortega, Javier García-Rubio, Alejandro Vaquera, Sergio J. Ibáñez

**Affiliations:** 1Faculty of Sports Science, University of Extremadura. Optimization of Training and Sport Performance Research Group, 10071 Cáceres, Spain; jagaru@unex.es; 2Faculty of Sports Science, University of Murcia, 30730 San Javier, Spain; 3Faculty of Sports Science, University of Leon, 24001 León, Spain; avaqj@unileon.es

**Keywords:** basketball referee, external training load, internal training load, competition

## Abstract

(1) *Background*: The use of advanced technology to study the energy demands of sport participants during actual sport competition is an important current research direction. The purpose of this study was to identify the physiological, internal, and external demands placed on basketball referees using ultra-wideband (UWB) technology, in relation to the period of the game. (2) *Methods:* The sample was comprised of nine international referees, and the data collection took place during the Women’s EuroBasket Sub-16 championship. Internal and external load were assessed through the inertial device WIMU PRO^TM,^ using UWB technology in order to quantify the effort exerted by each referee. The internal load was examined in relation to each individual’s heart rate (HR). The external load included the kinematic variables *accelerations (Acc), decelerations (Dec), Acc/min, Dec/min, distance covered, steps, maximum speed (Vmax), average speed (Vavg)*, and *speed zones,* as well as the neuromuscular variables *impacts (Imp), PlayerLoad^TM^ (PL^TM^)*, *PL^TM^/min*, *Metabolic Power (PMet)*, *and PMet/min.* (3) *Results:* The results exposed that referees work around 62% HRmax and spend more than 80% of the match at intensities between 0–12 km/h. The first period was the period in which the greatest work demand was experienced in relation to these neuromuscular outcomes (11.92 PL; 3.61 Met; 277 Impacts). The results revealed a diminishment of internal and external demands on the referees over the course of the game. (4) *Conclusions*: The results highlight the importance of monitoring and quantifying the workload of basketball officials, because doing so would allow for the establishment of individualized performance profiles that could be designed with the purpose of benefiting referee performance during games. The use of inertial devices allows for the objective quantification of referee workload under competitive circumstances.

## 1. Introduction

Basketball is a sport that has been in continuous evolution since its beginnings due to changes in the rules of the game that have been designed to create a more dynamic and attractive game for the spectator [1,2]. These changes have caused an exponential increase in the physical demands placed on the players and referees during competition [2]. Research that has been conducted to examine the workload demands of basketball participants has focused principally on players’ energy demands [3]; only infrequent studies have addressed the referees’ workloads. As such, it is necessary to better understand the demands of the game that the referees experience [4], as doing so should benefit referee physical preparation and self-efficacy [5]. This knowledge would also be vital for the development of individualized physical training programs [6], and to better understand the decision-making process as it occurs during competitive circumstances [7].

The physical demands of refereeing the sport of basketball include internal demands, most commonly determined through heart rate (HR) values, and external demands [8]. Various studies have assessed heart rate in referees of team sports, such as women’s basketball (60–70% HRmax) [9]; men’s basketball (~80% HRmáx) [10,11,12]; football (>80% HR) [13,14]; futsal (~80% HRmax) [15,16], and handball (60–70% HRmax) [17]. Other research studies have utilized additional work parameters focused instead on the kinematic and neuromuscular workload demands, such as distance covered, speed, accelerations, metabolic energy, or PlayerLoad **^TM^** [18,19,20,21,22]. These parameters provide additional objective data relative to the physical demands placed on referees during competitive circumstances: individual fitness level, type of competition (local, national, or international), numbers of referees (two or three), gender of teams (men or women), and characteristics of the game (close game, game style, etc.) [23]. In addition, these variables of internal and external load can be associated with the levels of stress and fatigue of referees during their intervention, which may affect referees’ decision making [24,25]. In order to develop a suitable decision-making process, it will be necessary to have an adequate fitness condition profile to be situated in the right place that enables accurately judging the actions that occur during the competition.

Included among the measurement methods that allow for the assessment of kinematic variables (distance covered, speed, range of speed) are pedometers [22], video-based tracking (VBT) [1,20], and global positioning systems such as the Global Navigation Satellite System (GNSS), which requires a tracking device (HDOP) and a quality signal (GDOP). The VBT system is a non-invasive system, given that the players and the referees do not need to apply any sort of electronic tracking device. Nonetheless, the use of this system is limited to spaces where video cameras are installed, and thus presents an indirect albeit reliable means of measurement. The GNSS technology requires that players wear a device that enables the tracking of the individual’s positioning through a satellite [26], and is thus limited to assessing referee outcomes in sports that are played outdoors, such as football [19,27,28] or rugby [21]. Nonetheless, with the development of the ultra-wideband (UWB) technology, it is possible to identify the positioning of athletes and referees in indoor sports with greater confidence through the GNSS technology [29,30,31]. Installation of the system is quite simple, and only requires the placement of six antennas to obtain radio frequencies that determine the nearly exact positioning of each athlete or official with only the slightest errors [28]. The use of these devices allowed the quantification of workload demands and performance in an objective manner, and enables the possibility of working more precisely with those behavioral outcomes that can be of greatest benefit to each referee [10].

Use of the UWB devices has been relatively recent [30,32], and their use has typically been limited to the examination of the validity and reliability of these instruments in experimental situations [29,31]. The present study had the purpose of determining the internal and external workload demands placed on basketball referees during a recent international competition, to subsequently establish training programs according to the needs of each referee. In addition, the study was focused on understanding variation in workload demand across periods of play. The UWB technology was employed to address these research interests. 

## 2. Materials and Methods

### 2.1. Design

The present study utilized a cross-sectional design with existing groups [33] to better understand the internal and external demands placed on referees at the international level. The study took place during the Women’s EuroBasket U-16 championship that was held in Gibraltar in 2017.

### 2.2. Participants

The sample was comprised of nine (six male and three female) international-level basketball referees. The characteristics of the sample are provided in Table 1. These referees officiated 15 games during the championship. The participants were appointed by the International Basketball Federation (FIBA). Therefore, it was not possible to conduct fitness tests before the championship.

### 2.3. Ethics

Each of the referees was informed of the experimental protocol and the possible benefits of their participation before providing their consent to participate in the study. Each of the participants provided a signed informed consent prior to involvement. Each of the procedures included in the study were approved in advance by the Ethical Committee of the University of Extremadura (nº17/2017), and in accordance with the Helsinki Declaration (2013). The FIBA also authorized all the protocols that were utilized.

### 2.4. Variables

The period of play served as the independent variable in this investigation. At this level of play, the game consists of four 10-minute periods (or quarters) of play. The dependent variables were each of the internal (physiological) demands and external (kinematic and neuromuscular) demands that were assessed in the study. Thus, the study was designed to determine the workload demands placed on basketball referees and to identify any differences that may have emerged in these workload demands over the four quarters of the game.

### 2.5. Internal Load

Heart rate was defined as the percentage of maximum heart rate (%HRmax) that was experienced by each referee in each of the four periods of the game. Six workload categories, or heart rate (HR) zones, were utilized that corresponded with those proposed by Vaquera et al*.,* [12]: *HR_1_* (50–60% HRmax); *HR_2_* (60–70% HRmax); *HR_3_* (70–80% HRmax); *HR_4_* (80–90% HRmax); *HR_5_* (90–95% HRmax); *HR_6_* (>95% HRmax). Each referee’s *HRmax* was established by determining the maximum HR value that they attained over the course of the game, and the heart rate zones reflected the percentage with each individual’s maximum heart rate. The *HRmax* for each referee was calculated in accordance of the formula proposed by Whaley, Kaminsky, Dwyer, Getchell, and Norton [34]:
(1)MenHRmax (bpm)=203.9−[0.812∗Age]+[0.276∗HRbasal]−[0.084∗Body Mass]
(2)WomenHRmax (bpm)=204.8−[0.718∗Age]+[0.162∗HRbasal]−[0.105∗Body Mass]

### 2.6. External Load

The external variables employed in this study were: accelerations (Acc); decelerations (Dec); distance covered (Dis); steps (St); maximum speed (Vmax); average speed (Vavg); impacts (n), PlayerLoad^TM^ (Pl) and Metabolic Power (Pmet).

Firstly, the number of *accelerations (Acc)* and *decelerations (Dec)* per game and period were determined, as well as *Acc* and *Dec* per minute in each period. The value established for its calculation was greater than 1 or −1 m/s^2^.

The variables of *distance covered (meters)* and *steps (n)* were measured through the inertial device, taking its values by periods and throughout the game.

The maximum speed (Vmax), the average speed (Vavg), and different speed zones were determined for the speed variable. The ranges of speed zones were: V_1_ = 0–6 km/h (walking); V_2_ = 6–12 km/h (jogging); V_3_ = 12–18 km/h (running); and V_4_ = 18–21 km/h (sprinting). These ranges of speed were established according to criteria developed by Gómez-Carmona and Pino-Ortega, [19] and based on the characteristic demands of basketball officiating.

*Impacts* were measured by G*-*forces that the referees experienced during different game actions. The calculation of this variable was achieved by summing the vector of G-forces that the referee experienced across three axes *(*x, y, and z). The impact value was established when the G-forces of the movement exceeded 5 Gs [35].

The variable of *PlayerLoad (PL*) was defined as the sum of the acceleration vectors as assessed through the inertial device employed (WIMU PRO^TM^) and in each of the three axes (vertical, anterior-posterior, and lateral). The *PlayerLoad* variable has demonstrated strong reliability and validity indices [36,37] to assess the neuromuscular load of each referee [38]. The corresponding value was calculated through the following equation, where *Z* represents acceleration from the anterior to posterior direction, *X* refers to acceleration in the medial-lateral direction, and *Y* refers to vertical acceleration. Time is represented by *t* and *n* refers to number.
(3)$$PlayerLoad_{t=n}\overset{t=n}{\underset{t=0}{\sum }}\sqrt{\frac{{\left(Z_{t=i+1}-\;Z_{t=i}\right)}^{2}+{\left(X_{t=i+1}-\;X_{t=i}\right)}^{2}+{\left(Y_{t=i+1}-Y_{t=i}\right)}^{2}}{100}}$$

The variable of *Metabolic Power (PMet)* was obtained by multiplying the velocity (*V*) by the energy cost of the activity (*EC*), as represented by the rate of acceleration [39]. The value was obtained through the following formula:(4)PMet=EC · V

### 2.7. Materials

A variety of instruments were used in the study. The Tanita Model BC-601 (TANITA, Tokyo, Japan) scale was used for the measurement of body composition values. A portable tallimeter device (SECA, Hamburg, Germany) was used to assess the height of each referee, and each individual was measured without shoes. To determine internal demand (%HR), the GARMIN^®^ Cardiac Frequency bands were used. This instrument consists of a belt with two electrodes and a small electronic device that receives the signals from the electrodes; then, the received data is synchronized with the inertial device WIMU PRO^TM^ (Realtrack System, Almería, Spain) through wireless ANT^+^ technology [40]. The WIMU PRO^TM^ (wireless inertial unit movement) was used to monitor the external and internal loads. The device was integrated through different sensors (global position system, or GPS), three accelerometers, a gyroscope and an ultra-wideband (UWB) chip. The frequency for the chip’s signal was 18 Hz. The UWB-based system has been valid and reliable for analysis in outdoor and indoor sports [41]. In addition, a six-radiofrequency antenna system was used, which was located near the floor. This system allowed the location of each referee to be known at each moment. The calibration of these antennas was made according to the dimensions of the court, as reflected in Figure 1. Then, the data that were obtained through the devices were analyzed through specific software S-PRO^TM^ (RealTrack Systems, Almería, Spain). All appropriate permissions have been obtained for WIMU PRO systems and S-PRO software for research purposes, as well as mention in subsequent publications.

### 2.8. Statistical analysis

The first step in the statistical analysis involved a test of parametric assumptions. These tests were conducted to examine homoscedasticity, the presence of outliers, and the normality assumption [42]. The results supported the suitability of the data in meeting the parametric assumptions and subsequent parametric analysis. A descriptive analysis of the data was conducted, and calculations of each variable’s mean value, standard deviation, associated confidence intervals, and minimum and maximum values were conducted. Finally, a one-factor analysis of variance (ANOVA) was conducted to determine if differences in referee workload levels varied by game period. In this analysis, *effect size (f) and power (ϕ)* were also calculated. For the interpretation of these findings, effect sizes were established as follows: *f = 0.10* small; *f = 0.25* medium*;* and *f = 0.40* large [43]. Bonferroni post-hoc analysis was used to identify any significant differences in workload outcomes across periods. The statistical analysis was completed through statistical package SPSS (SPSS Inc., Chicago IL, EEUU) version 21. Significance criteria were established throughout at *p < 0.05.*

## 3. Results

Descriptive and inferential data of the internal load are presented in Table 2. The heart rate data indicates that these referees worked above 62% *HRmax* with a *HRavg* of 124 beats per minute and spent 83% of each period between 50–80% *HRmax*. The analysis was conducted through a single-factor ANOVA with a subsequent post-hoc Bonferroni analysis revealing significant differences between the second and third periods in %HRmax (*p* ≤ 0.05) and HRavg (*p* ≤ 0.05). Significant differences also emerged between the first and third periods in *70–80% HRmax* (*p* ≤ 0.05).

Table 3 and Table 4 provide the results of the study in relation to the external variables. It can be noted that the referees engaged in roughly 270 accelerations and 128 decelerations per period. They covered an average of 4330 m per game, reaching a peak velocity of 18.91 km/h with an average velocity of 4.87 km/h while taking 2616 steps per game. During 80.43% of game time, the referees worked at a velocity between 0–12 km/h. The referees received 1025 impacts on average per game, which reflects a workload of 44.13 PL and an energy cost of 15.60 Mets. The first period was the period in which the greatest work demand was experienced in relation to these neuromuscular outcomes (11.92 PL; 3.61 Met; 277 Impacts). The same tables provide the results of the analysis that compared differences across periods. In relation to this, a single-factor ANOVA with a subsequent post-hoc Bonferroni analysis revealed that significant differences existed between the first and third periods in *average speed (p ≤ 0.05)* and amount of time spent in the *speed zone V_3_ (p ≤ 0.01).* Significant differences also emerged between the first and fourth quarter in average speed *(p ≤ 0.01); steps taken (p ≤ 0.05); PlayerLoad (p ≤ 0.01); PlayerLoad/min (p ≤ 0.05); Metabolic Power (p ≤ 0.01); Metabolic Power/min (p ≤ 0.05); distance covered (p ≤ 0.01); distance covered/min (p ≤ 0.05); speed zones V_3_ (p ≤ 0.01); and V_4_ (p ≤ 0.01).* Finally, significant differences were found in *speed zone V_3_* between the first and third periods *(p ≤ 0.05)*, as well as between the second and fourth periods in the same speed range *(p ≤ 0.05).* The effect sizes were large for each of these differences, and were as follows: *Vavg (f = 0.997); Steps (f = 0.961); PL (f = 0.973); PL/min (f = 0.973); PMet (f = 0.969); PMet/min (f = 0.976); Distance (f = 0.982); Distance/min (f = 0.986); V_3_ (f = 0.925);* and *V_4_ (f = 0.614)*.

## 4. Discussion

The purpose of this study was to identify the internal and external demands placed on basketball referees through UWB technology and in relation to the period of the game. The results highlight the need to monitor and quantify the various types of demands placed on these officials in order to establish a performance profile that will facilitate interventions with referees to benefit their performance in competition [4,6]. The principal findings from this study revealed a decrease in workload demands over the course of the game.

Heart rate levels have been previously examined as an indicator of internal load on referees. Previous research has found values corresponding to 60–70% *HRmax* as being customary in handball [17] and basketball [9], and for values to exceed 80% in futsal [15] and in a different basketball study [11,12]. In the present study, the referees were working around 62*% HRMax*, which were similar values to those obtained in previous studies [9], but far below values in recent studies (>85*% HRMax*) [11,12]. However, heart rate values can vary in relation to the type of officiating, the level of competition, and the gender of the players [12,23], as well as the ability level of the players. In fact, the only study in which researchers found a *%HRMax* similar to this study was in a women’s tournament [9], as in the present investigation. In men’s basketball, the game is played at a greater tempo, owing to the greater relative physiological capacities of the players [44]. In this study, the *%HRMax* was calculated in relation to the *HRmax* of each referee in each period of play through current and direct measurement. In other studies, *HRMax* was determined in a more indirect manner [9,11], thus generating a less precise assessment, which makes it difficult to compare findings directly with the present data. It is recognized that elite level athletes generally make good decisions under circumstances of stress and fatigue [45], particularly in those circumstances of great challenge [46]. The referees in this study were not generally in a state of high intensity. The decision-making quality tends to decrease in circumstances of moderate intensity for many participants in sport, which makes it more probable that referees will make mistakes at moderate and low physical demand [11]. Low intensity is associated with the greatest likelihood of committing mistakes, but it should be noted that *HRmax* is also reflective of the gender of the players participating [12]. As such, it would be necessary to conduct studies that examine how decision-making quality in referees varies in relation to physiological stress and accumulated fatigue during games with the goal of preparing accordingly for the circumstances that will be encountered by the referees, particularly for the demands at the end of games, since each decision has a great impact on the results [25].

In relation to the external variables, it is clear that the findings from this study are consistent with previous results relative to distance covered by referees in similar research studies (around 4400 m per game) [1,20,22], even though the instruments that were used to obtain the data were less direct, such as pedometers or VTB. The UWB technology was utilized in this study as a means of collecting data, and this methodology has greater validity than the VTB approach [31], as well as greater precision and better validity and reliability estimates than the distance and speed data that is acquired through GPS technology [29]. In terms of the running velocity of the referees in this sample, the findings are consistent with results obtained in other studies with basketball referees [1]. In this case, referees spend ~80% of the game time in the walking and jogging (0–12 km/h) speed zone, which probably has matches with low intensity, depending on the game style and the characteristics of the competition, teams, and players. Instead, football referees perform movements below 12 km/h for 65% of the match [19]; these results are different because the distance covered in football is more than basketball. In both cases, similar time values exist in high intensity or sprint actions. Accelerations and decelerations can also be determined through changes in direction and rhythm in team sports in response to the opponent’s actions relative to the field of play [47]. The acceleration and deceleration data is also of vital importance for referees given their need to match the demands placed upon the players during the game with the constant changes of direction that occur as players shift direction from one basket to the other and move side-to-side in accordance with the movement of the ball. To the best of our knowledge, this is the first study to examine acceleration and deceleration in basketball referees in a game situation. The results revealed a total of 1079 accelerations and 505 decelerations per game. High-intensity and low-intensity accelerations and decelerations have also been linked to markers of creatine kinase (CK) and musculoskeletal damage, given that they are actions that the body cannot readily absorb [37]. This muscular damage negatively affects muscular function over the subsequent 24 h and is still perceived as muscular soreness by athletes an additional 24 h later [48]. Consequently, monitoring this information may permit better scheduling of rest for referees, especially in condensed championship circumstances in which there are games played every day, as was the case in the present study.

The data from others variables, such as the PL, allow for an improved understanding of the demands and volume of effort experienced by the referees during the game, which allows for a better determination of referee performance outcomes. Recent studies have found that these types of variables constitute the best type of data for predicting performance in athletes [3,49], and are the indicators most strongly associated with speed, acceleration, and deceleration [50]. Unconventional movements, such as lateral or side-to-side movements and backwards running, are movements carried out at low speed, but which place high demand on athletes. These actions are quantified through the PL. Other measurement parameters for internal and external demands may underestimate the workload experienced by athletes and, as such, may not be reliable indicators to use in training, and additional accelerometry data may be needed to supplement this information [51]. The PL data is also useful because it provides information relative to the workload that athletes experience during training and competition, and is related to the risk of injury in players [37]. The use of Metabolic Power allows for the development of workload profiles in training and competition in team sports [52], and reflects the energy expenditure of referees, given the strong relationship between aerobic capacity and oxygen consumption [39,50].

The findings from this study reveal that cardiovascular stress decreases over the course of the game, which is consistent with previous research [53]; the study also revealed the external demands such as distance covered, PL, and PMet. Various studies indicate that this occurrence may be due to the accumulated fatigue of the referees or due to the decrease in the pace of play of the game as players become fatigued [54]. However, the previous research was all conducted in football, where there are a limited number of substitutions permitted, and the majority of players remain on the pitch for the entire game, which contributes to greater accumulated fatigue among the players. In the sport of basketball, the players may be substituted, and then return to play, following rest periods during the game. In addition, the greatest interruptions to play occur during the final five minutes of the game [55]. Although the end of the game is associated with the greatest potential for change in the game’s outcome [56], the frequent interruptions to play allow for the physical recuperation of the referees and the players during this time frame. To the contrary, the beginning of the game is associated with the time frame in which teams try to impose their style of play on the opponent with the objective of gaining an early advantage that will translate to a lead on the scoreboard [56]. Thus, it is necessary to understand the dynamics with which the games are played in order to best interpret and quantify the variables that influence the performance of referees. Furthermore, it is essential to adjust the preparation of the referees according to the specific demands that are anticipated in relation to each period of play in order to minimize the decision-making errors that may occur due to sub-optimal physical preparation practices.

## 5. Conclusions

The findings permit some conclusions relative to internal and external outcomes through use of the inertial device (WIMU PRO^TM^), which enables an objective and reliable assessment of the workload demands experienced by basketball referees during competition. These findings are only determined for the sample that has participated in this study. It would be interesting to continue in this way with referees of different categories and competitions. In addition, monitoring and quantifying the workload of basketball referees is necessary to design individualized training programs for each official. Such training programs should be adjusted according to specific competitive characteristics to conform to these demands.

## 6. Practical Applications

It is also recommended that integrated training programs be conducted that combine physical demands with decision-making demands in the natural environment of officiating. For example, training programs can be designed where there is an intense workload at the beginning with elevated *HRmax*, or long training sessions in which there is considerable accumulated fatigue, but at a work demand that does not surpass 60–70% of *HRmax* while engaging in game-related decision-making. These different approaches to physical preparation should provide referees with the opportunity to perform at a higher level during critical moments of the game.

## Figures and Tables

**Figure 1 ijerph-16-03421-f001:**
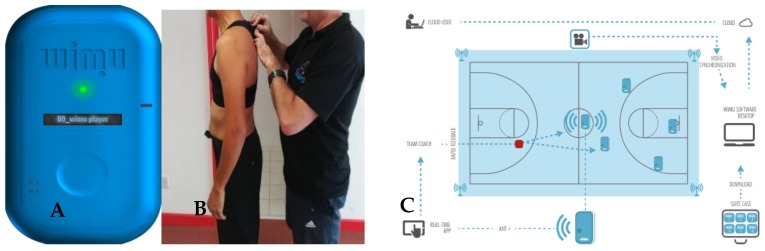
A: WIMU PRO^TM^ (wireless inertial unit movement) ready to use; B: Introduction of the WIMU PRO device on a specific harness; C: The antenna system’s placement (All appropriate permissions have been obtained from the copyright holders as well as written informed consent from the depicted individuals for the publication of this image).

**Table 1 ijerph-16-03421-t001:** Anthropometric and demographic data of referees (N = 9).

	Male (6)			Female (3)		
	Mean	Minimum	Maximum	Mean	Minimum	Maximum
**Age (Yrs.)**	44	37	49	39	34	44
**Experience (Yrs.)**	25	20	32	15	10	21
**Height (m)**	1.83	1.78	1.96	1.74	1.70	1.80
**Body Mass (kg)**	78.48	67.3	92.5	71.13	61.2	86.9
**Body Fat (%)**	13.55	9.5	16.3	23.87	19.9	29.2
**Muscle Mass (%)**	64.27	55.7	73.6	50.80	46.5	57.9
**Bone Mass (%)**	3.33	2.9	3.8	2.73	2.5	3.1
**BMI**	23.17	20.4	25.1	23.27	20.4	26.8
**DCI (kcal)**	4079	3530	4748	2865	2619	3284
**Metabolic Age (Yrs.)**	29	22	34	27	19	37
**Body Water (%)**	62.28	59.5	65.5	55.67	51.6	58.4
**Visceral Fat (%)**	5	4	7	3	1	5

Note: BMI = Body Mass Index; DCI = Daily Calorie Intake.

**Table 2 ijerph-16-03421-t002:** Descriptive and Inferential Results of the Internal Load Variables.

Variables		Periods	Mean	SD	95% IC		Min	Max	*F*	*Sig.*	*_f_*	*ϕ*	*p*	DbP
					L	U								
***Internal Load Variables***														
**%HRMax**		1	63	8	61	65	42	77	3.155	0.026	0.982	1.000	*	π
		2	64	8	61	66	47	82						
		3	59	9	57	62	36	85						
		4	60	9	57	63	36	86						
		TG	62	7	60	64	47	78						
**HRavg (bpm)**		1	126	15	122	131	84	154	3.155	0.026	0.982	1.000	*	π
		2	128	17	123	133	93	164						
		3	119	18	113	124	71	169						
		4	120	18	114	125	72	172						
		TG	124	14	119	128	93	156						
**Zones HRMax (% time per zones)**	**50–60%**	1	25.52	21.00	19.14	31.91	0.00	73.22	2.505	0.061	0.669	1.000		
		2	26.83	21.64	20.25	33.41	0.00	69.67						
		3	35.51	21.09	29.17	41.84	0.00	65.12						
		4	34.61	23.31	27.61	41.62	0.00	69.73						
		TG	30.93	18.10	25.50	36.37	0.00	64.57						
	**60–70%**	1	40.25	21.76	33.63	46.87	0.00	75.76	0.814	0.488	0.741	1.000		
		2	38.74	23.06	31.72	45.75	0.00	81.87						
		3	33.38	20.73	27.15	39.60	0.00	74.45						
		4	36.32	23.25	29.34	43.31	0.00	88.24						
		TG	36.88	19.39	31.05	42.70	0.00	72.41						
	**70–80%**	1	20.13	22.54	13.27	26.98	0.00	76.90	2.799	0.042	0.379	1.000	*	β
		2	18.68	20.87	12.34	25.03	0.00	71.83						
		3	10.53	17.21	5.36	15.70	0.00	70.30						
		4	11.37	17.59	6.08	16.65	0.00	84.37						
		TG	15.04	17.89	9.67	20.42	0.00	75.35						
	**80–90%**	1	2.91	8.42	0.35	5.47	0.00	51.50	0.928	0.428	0.091	0.986		
		2	4.29	11.06	0.92	7.65	0.00	51.44						
		3	1.70	7.08	–0.42	3.83	0.00	42.79						
		4	1.71	6.78	–0.33	3.75	0.00	36.07						
		TG	2.65	6.09	0.82	4.48	0.00	27.50						
	**90–95%**	1	0.28	1.88	–0.29	0.86	0.00	12.48	0.325	0.807	0.033	0.673		
		2	0.70	2.68	–0.11	1.52	0.00	12.64						
		3	0.35	2.32	–0.35	1.04	0.00	15.59						
		4	0.32	2.12	–0.32	0.95	0.00	14.23						
		TG	0.40	1.49	–0.05	0.85	0.00	7.44						
	**>95%**	1	0.09	0.59	–0.09	0.27	0.00	3.93	0.479	0.697	0.015	0.364		
		2	0.19	1.26	–0.19	0.57	0.00	8.37						
		3	0.00	0.00	0.00	0.00	0.00	0.00						
		4	0.08	0.57	–0.09	0.26	0.00	3.81						
		TG	0.09	0.36	–0.02	0.20	0.00	2.04						

Note: SD = Standard Deviation; IC = Interval confidence; L = Low; U = Upper; Min = Minimum; Max = Maximum; DbP = Differences between periods; HRmax = Heart Rate Maximum; HRavg = Heart Rate Average; ** = *p* < 0.01;* = *p* < 0.05; f = Effect Size; ϕ = Power; β = Differences between first and third periods; π = Differences between second and third periods.

**Table 3 ijerph-16-03421-t003:** Descriptive and Inferential Results of the External Load Variables (Kinematic Variables).

Variables		Periods	Mean	SD	95% IC		Min	Max	*F*	*Sig.*	*_f_*	*ϕ*	*p*	DbP	
					L	U									
***Kinematic External Load Variables***															
**Accelerations (n)**		1	282.45	44.029	269.07	295.84	180	396	1.098	0.351	0.968	1.000			
		2	267.89	43.951	254.52	281.25	183	364							
		3	272.69	44.349	259.36	286.01	188	414							
		4	264.38	63.805	245.21	283.55	135	490							
		TG	1079.36	192.344	1021.57	1137.14	371	1639							
**Decelerations (n)**		1	131.50	39.032	119.63	143.37	58	236	0.246	0.864	0.920	1.000			
		2	126.91	35.824	116.02	137.80	70	208							
		3	126.16	32.891	116.27	136.04	61	227							
		4	125.07	43.434	112.02	138.12	53	270							
		TG	505.49	138.441	463.90	547.08	126	890							
**Acc/min (m/s^2^)**		1	13.99	2.16	13.33	14.64	9.86	17.80	0.589	0.623	0.977	1.000			
		2	13.70	2.07	13.07	14.33	10.25	18.40							
		3	13.72	2.18	13.06	14.37	9.58	19.67							
		4	13.39	2.15	12.74	14.03	9.44	17.98							
		TG	13.65	1.92	13.07	14.23	10.12	17.35							
**Dec/min (m/s^2^)**		1	6.56	2.13	5.91	7.21	3.23	11.46	0.158	0.924	0.921	1.000			
		2	6.53	1.91	5.95	7.11	3.54	11.98							
		3	6.38	1.78	5.85	6.92	3.11	11.20							
		4	6.32	1.82	5.78	6.87	3.14	10.40							
		TG	6.40	1.68	5.90	6.91	3.44	9.87							
**VMax (m/s)**		1	18.61	1.24	18.24	18.99	16.00	21.77	0.961	0.413	0.996	1.000			
		2	18.69	1.12	18.35	19.03	16.95	22.89							
		3	18.59	1.31	18.19	18.98	15.54	21.85							
		4	18.29	1.15	17.94	18.63	16.43	22.54							
		TG	18.91	1.17	18.56	19.26	16.86	22.89							
**Vavg (m/s)**		1	4.99	0.23	4.92	5.06	4.52	5.57	4.843	0.003	0.997	1.000	**	γβ	
		2	4.89	0.29	4.80	4.97	4.32	5.77							
		3	4.83	0.27	4.75	4.91	4.27	5.40							
		4	4.78	0.31	4.69	4.87	4.12	5.36							
		TG	4.87	0.21	4.80	4.93	4.45	5.48							
**Steps (n)**		1	713	130	673	752	389	979	3.504	0.017	0.961	1.000	*	γ	
		2	657	117	621	692	421	932							
		3	647	132	607	686	295	993							
		4	623	157	576	671	288	986							
		TG	2616	543	2452	2779	1065	3728							
**Steps/min**		1	35	6	33	37	22	49	2.093	0.103	0.961	1.000			
		2	34	7	32	36	21	47							
		3	33	7	31	35	13	44							
		4	32	7	30	34	12	46							
		TG	33	6	31	35	16	44							
**Distance (m)**		1	1164	145	1120	1208	865	1434	4.774	0.003	0.982	1.000	**	γ	
		2	1089	112	1055	1123	902	1339							
		3	1083	127	1045	1121	730	1309							
		4	1049	194	990	1107	690	1738							
		TG	4330	603	4149	4512	1832	5553							
**Distance/min (m/min)**		1	58	7	56	60	44	74	3.264	0.023	0.986	1.000	*	γ	
		2	56	7	54	58	45	78							
		3	54	6	53	56	44	71							
		4	54	7	51	56	42	68							
		TG	55	5	53	57	45	70							
**Speed Zones** **(m)**	**V_1_**	1	513.89	83.25	488.58	539.20	358.02	718.21	0.312	0.816	0.974	1.000			
		2	499.80	67.48	479.29	520.32	382.38	672.81							
		3	502.95	73.09	480.99	524.91	361.80	690.82							
		4	498.61	102.78	467.73	529.49	262.55	845.39							
		TG	2000.06	317.48	1904.68	2095.45	765.11	2777.68							
	**V_2_**	1	389.45	54.72	372.81	406.08	280.06	530.40	1.460	0.227	0.978	1.000			
		2	370.14	45.50	356.31	383.98	277.85	484.52							
		3	368.38	46.63	354.37	382.39	275.89	491.62							
		4	368.67	73.35	346.64	390.71	224.66	641.77							
		TG	1483.23	201.53	1422.69	1543.78	737.59	1901.84							
	**V_3_**	1	235.71	55.92	218.71	252.71	138.31	370.24	9.974	0.000	0.925	1.000	**	αβγε	
		2	202.40	61.41	183.73	221.07	86.73	399.46							
		3	194.79	62.52	176.00	213.57	52.88	329.70							
		4	169.15	51.16	153.78	184.52	69.67	339.18							
		TG	794.13	201.76	733.52	854.75	329.51	1399.68							
	**V_4_**	1	19.47	13.88	15.25	23.68	2.99	59.76	6.167	0.001	0.614	1.000	**	γ	
		2	13.99	10.93	10.67	17.31	0.00	45.15							
		3	14.89	12.02	11.28	18.50	0.00	53.39							
		4	9.03	8.54	6.46	11.60	0.00	33.16							
		TG	56.70	34.11	46.46	66.95	0.00	143.31							

Note: SD = Standard Deviation; CI = Confidence Interval; L = Lower; U = Upper; Min = Minimum; Max = Maximum; DbP = Differences between periods; Acc = Accelerations; Dec = Decelerations; Vmax = Maximum Speed; Vavg = Average Speed; ** = *p* < 0.01;* = *p* < 0.05; f = Effect Size; ϕ = Power; α = Differences between first and second periods; β = Differences between first and third periods; γ = Differences between first and fourth periods; ε = Differences between second and fourth periods.

**Table 4 ijerph-16-03421-t004:** Descriptive and Inferential Results of the External Load Variables (Neuromuscular Variables).

Variables	Periods	Mean	SD	95% IC		Min	Max	*F*	*Sig.*	*_f_*	*ϕ*	*p*	DbP
				L	U								
***Neuromuscular External Load Variables***													
**PlayerLoad (PL)**	1	11.92	1.89	11.35	12.50	8.58	16.03	4.940	0.003	0.973	1.000	**	γ
	2	11.08	1.75	10.55	11.62	7.81	14.64						
	3	11.04	1.78	10.50	11.58	6.98	14.47						
	4	10.39	2.10	9.76	11.02	7.08	14.91						
	TG	44.13	7.92	41.75	46.51	19.06	62.77						
**PlayerLoad/min (PL/m)**	1	0.59	0.09	0.56	0.62	0.40	0.86	3.134	0.027	0.973	1.000	*	γ
	2	0.57	0.10	0.54	0.60	0.36	0.82						
	3	0.55	0.09	0.53	0.58	0.39	0.85						
	4	0.53	0.10	0.50	0.56	0.37	0.79						
	TG	0.56	0.08	0.53	0.58	0.39	0.77						
**Power Metabolic (PMet)**	1	3.61	0.48	3.47	3.76	2.54	4.60	4.615	0.004	0.969	1.000	**	γ
	2	3.43	0.93	3.15	3.71	2.69	9.03						
	3	3.32	0.40	3.20	3.45	2.15	4.25						
	4	3.14	0.51	2.99	3.29	2.11	4.74						
	TG	15.60	2.57	14.83	16.37	5.60	21.65						
**Power Metabolic/min (PMet/m)**	1	0.18	0.02	0.17	0.19	0.14	0.23	4.104	0.008	0.976	1.000	**	γ
	2	0.18	0.04	0.16	0.19	0.14	0.38						
	3	0.17	0.02	0.16	0.17	0.13	0.22						
	4	0.16	0.02	0.15	0.17	0.10	0.20						
	TG	0.20	0.03	0.19	0.21	0.15	0.27						
**Impacts**	1	277	100	247	307	84	542	1.450	0.230	0.883	1.000		
	2	257	99	227	287	76	480						
	3	262	91	234	289	98	445						
	4	236	90	209	263	86	418						
	TG	1025	372	913	1136	344	1890						

Note: SD = Standard Deviation; CI = Confidence Interval; L = Low; U = Upper; Min = Minimum; Max = Maximum; DbP = Differences between periods; ** = *p* < 0.01; * = *p* < 0.05; f = Effect Size; ϕ = Power; γ = Differences between first and fourth periods.

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
