# Peer review of "Internal and External Demands in Basketball Referees during the U-16 European Women’s Championship"

_ijerph, 2019, doi:10.3390/ijerph16183421_

Round 1
Reviewer 1 Report
This is a well-presented paper on an interesting area of sport officiating.
While you have conducted a rigorous set of physiological tests on the participants, it is pleasing to note the attention paid thereafter to the associated cognitive/psychological demands, i.e., accuracy in decision-making under game conditions, particularly at the 'business end' of the contest.
Please find attached some comments/suggestions re very minor edits to the main text.

Author Response
Dear Reviewer,
I enclose the document with the corrections made after its review. All contributions have been taken into account. The changes made are indicated in red.
Thanks for you time and dedication

Reviewer 2 Report
General Comments:
The authors have provided a detailed assessment of the demands of basketball refereeing. The findings of the study will be beneficial to the design and development of physical preparation training programs for sports officials in basketball and similar sports. I commend the authors on the innovative approach they have taken to measuring the numerous variables and the detailed analysis they have completed. To further enhance the manuscript however, I would believe the introduction needs to provide a bigger "why" for the importance of the study. Please be more explicit regarding the benefits of this knowledge. I would also like to see some further reference to the impact of fatigue on decision making in the introduction. Also is there any background information on the participants training history? The individual fitness level of each referee may have a considerable impact on the findings of this study. Similarly would the game scenario influence the findings (i.e. is there differences between a close fourth quarter game and a game the one side has dominated)?
I believe more depth could be added to the discussions. In particular how the findings relate to the previous research. Also, while it may be outside the scope of this study, were any performance indicators assess to identify the impact of these demands on officials? You make reference to decision making but do not appear to assess this or integrate the literature on this. One study is noted on this topic but I would suggest there are others. For example: Larkin et al. (2014) Assessment of decision-making performance and in-game physical exertion of Australian football umpires, Journal of Sports Sciences; Paradis, Larkin and O'Connor (2016) The effects of physical exertion on decision-making performance of Australian football umpires, Journal of Sport Sciences; and Helsen and Bultynck (2004) Physical and perceptual-cognitive demands of top-class refereeing in association football, Journal of Sport Sciences.
Specific Comments:
L35-36 - Please add reference to the sentence 'These changes have caused...'
L45-46 - Can the authors provide an overview of the findings from these studies
L47 - Should 'distance recovered' be 'distance covered'?
L49 - Again can an overview of the findings from these studies be included
DISCUSSION
L7 - 'decrease in workload demands' are the authors certain this is not a fitness related issue?
L11-12 - the authors state 'these referees were working was 62%, or much below the values obtained in other studies' but in L9 the authors have stated that basketball referees have values of 60-70%. Please revise.
L14-15 - One study has looked at the demands of referees in women's basketball, much greater reference is required on how the present study relates
L68 - what impact does game scenario have?
L77-78 - this is a unique feature of basketball and more discussion is required related to the frequent pauses in game play and the ability to aid recovery
Author Response
Dear Reviewer,
The introduction needs to provide a bigger "why" for the importance of the study. Please be more explicit regarding the benefits of this knowledge. Thank you very much for your suggestion. It has developed a more detailed explanation of the importance of the study.
I would also like to see some further reference to the impact of fatigue on decision making in the introduction. Thank you for your input. It has taken into account the inclusion of new recommended in relation to fatigue in decision-making studies.
Also is there any background information on the participants training history? Thank you for your comment. The reason for not being able to have a previous history of the participants has been specified.
I believe more depth could be added to the discussions. In particular how the findings relate to the previous research. Also, while it may be outside the scope of this study, were any performance indicators assess to identify the impact of these demands on officials? You make reference to decision making but do not appear to assess this or integrate the literature on this. One study is noted on this topic but I would suggest there are others. For example: Larkin et al. (2014) Assessment of decision-making performance and in-game physical exertion of Australian football umpires, Journal of Sports Sciences; Paradis, Larkin and O'Connor (2016) The effects of physical exertion on decision-making performance of Australian football umpires, Journal of Sport Sciences; and Helsen and Bultynck (2004) Physical and perceptual-cognitive demands of top-class refereeing in association football, Journal of Sport Sciences. Thank you very much for your recommendation. The previous data of other investigations have been specified. In addition, the recommended articles in relation to decision making have been added.
Specific Comments:
L35-36 - Please add reference to the sentence 'These changes have caused...'. Thanks for the suggestion. A reference to this comment has been specified.
L45-46 - Can the authors provide an overview of the findings from these studies. Thanks for the suggestion. The most relevant data from previous studies have been detailed.
L47 - Should 'distance recovered' be 'distance covered'? Thanks for your comment. One expression has been modified by another.
L49 - Again can an overview of the findings from these studies be included. Thank you for your contribution. The values of these studies have been specified.
Discussion
L7 - 'decrease in workload demands' are the authors certain this is not a fitness related issue? Thanks for your suggestion. The comment made has been taken into account, but the authors consider the finding made adequate.
L11-12 - the authors state 'these referees were working was 62%, or much below the values obtained in other studies' but in L9 the authors have stated that basketball referees have values of 60-70%. Please revise. Thanks for your recommendation. The explanation of these findings has been rewritten for better understanding.
L14-15 - One study has looked at the demands of referees in women's basketball, much greater reference is required on how the present study relates. Thank you for your input. It has been specified as it has been done in the present study
L68 - what impact does game scenario have? Thanks for your suggestion. But it has not been understood what the reviewer means by this matter.
L77-78 - this is a unique feature of basketball and more discussion is required related to the frequent pauses in game play and the ability to aid recovery. Thank you for your comment. It has been determined that it is well specified about the characteristics of Basketball.
Thanks for you time and dedication
Round 2
Reviewer 2 Report
The authors have addressed the majority of concerns well and have now presented a more insightful manuscript, which adds to the current literature and previous new information in relation to the physical preparation of sports officials (while also offering considered suggestions for future research).